# Titania supported synergistic palladium single atoms and nanoparticles for room temperature ketone and aldehydes hydrogenation

Long Kuai[1,2], Zheng Chen [3], Shoujie Liu[1], Erjie Kan[2], Nan Yu[1], Yiming Ren[2], Caihong Fang[1], Xingyang Li[2], Yadong Li[3]* & Baoyou Geng[1]*

Selective reduction of ketone/aldehydes to alcohols is of great importance in green chemistry and chemical engineering. Highly efficient catalysts are still demanded to work under mild conditions, especially at room temperature. Here we present a synergistic function of single-atom palladium ($Pd_1$) and nanoparticles ($Pd_{NPs}$) on $TiO_2$ for highly efficient ketone/aldehydes hydrogenation to alcohols at room temperature. Compared to simple but inferior $Pd_1/TiO_2$ and $Pd_{NPs}/TiO_2$ catalysts, more than twice activity enhancement is achieved with the $Pd_{1+NPs}/TiO_2$ catalyst that integrates both $Pd_1$ and Pd NPs on mesoporous $TiO_2$ supports, obtained by a simple but large-scaled spray pyrolysis route. The synergistic function of $Pd_1$ and $Pd_{NPs}$ is assigned so that the partial $Pd_1$ dispersion contributes enough sites for the activation of C=O group while $Pd_{NPs}$ site boosts the dissociation of $H_2$ molecules to H atoms. This work not only contributes a superior catalyst for ketone/aldehydes hydrogenation, but also deepens the knowledge on their hydrogenation mechanism and guides people to engineer the catalytic behaviors as needed.

---

[1] College of Chemistry and Materials Science, The Key Laboratory of Functional Molecular Solids, Ministry of Education, Anhui Laboratory of Molecular-Based Materials, The Key Laboratory of Electrochemical Clean Energy of Anhui Higher Education Institutes, Anhui Normal University, No. 189 South Jiuhua Road, Wuhu 241002, P. R. China. [2] School of Chemical and Biological Engineering, Anhui Polytechnic University, Beijing Middle Road, Wuhu 241000, China. [3] Department of Chemistry, Tsinghua University, Beijing 100084, China. *email: ydli@tsinghua.edu.cn; bygeng@mail.ahnu.edu.cn

Selective reduction of ketone/aldehydes to alcohols constitutes a great importance in green chemistry and chemical engineering[1,2]. The conventional noncatalytic method uses stoichiometric reducing agent (e.g., $NaBH_4$) and derives a large amount of inorganic wastes. In this regard, the catalytic hydrogenation is more environmentally friendly and atomically economical because the reductant is abundant $H_2$ and the ideal by-product is just renewable $H_2O$[3–6]. As we known, the Pd/C and Raney Ni are the most used commercial catalysts for industry hydrogenation reactions. While, Raney Ni as well as other Pd-free catalysts are not often found in the hydrogenation of ketone/aldehydes because harsh conditions (high temperature and $H_2$ pressure) are demanded to acquire high reaction kinetics[7,8]. Besides, the selectivity to alcohol presents low level under harsh hydrogenation conditions[5,7,8]. As for Pd/C catalysts, the active sites are Pd nanoparticles (NPs). Although they can be used under mild conditions, the catalytic efficiency are rather low due to the limited metal dispersion of Pd NPs[9–11]. Typically in the recent work of Zheng' group, we can find that the commercial Pd/C still exhibit quite low reactivity in the case study of the hydrogenation of benzaldehyde[11]. Thus, it is still a challenging work to develop efficient catalysts for selective ketone/aldehydes hydrogenation under mild conditions, especially at room temperature.

In terms of ketone/aldehydes hydrogenation, both the activation of C=O group and dissociation of $H_2$ are essential to the reaction kinetics[12,13]. Just as such, ketone/aldehydes hydrogenation greatly challenges both the conventional nanocatalysts and the recently focused single-atom catalysts. It is well known that the noble metal nanocatalysts (2–5 nm) such as Pd, Pt are active to ketone/aldehydes hydrogenation[14,15], but they are suffered from not only the cost and Earth-scarcity but also low atomic efficiency. So, the reported noble metal nanocatalysts did not present satisfactory reactivity. To address the problems of noble metal nanocatalysts, in the recent years, great efforts have been focused on downsizing the supported nanocrystals to single-atom dispersed active sites for acquiring 100% atomic efficiency[16,17], which has been greatly successful toward widely reactions such as CO oxidation[18], water-gas shift reaction[19,20], $O_2$ electro-catalysis[21], $CO_2$ electroreduction[22], etc. Unfortunately, it does not work for ketone/aldehydes hydrogenation because it damages the activity seriously. In the single-atom sites, the dissociation of $H_2$ has to go heterolytic pathway, whose barrier is much higher than that in the nanosized metal surface with homolytic dissociation, especially the single-atom sites are strongly pre-covered by the substrate[23,24]. Thus, in the hydrogenation field, it can be understood that the single-atom noble metal catalysts are mostly presented in the selective semi-hydrogenation[23,25–27], or show routine activity for aldehydes hydrogenation under mild conditions[11]. Therefore, it is emergency to design a single-atom based active sites that maintains high atomic efficiency but without sacrifice of the capacity for $H_2$ dissociation toward ketone/aldehydes hydrogenation. Hydrogen spillover is a well-known phenomenon in catalysis[28–30]. The spillover H species are high-active to reduce $Ti^{4+}$ for preparing black $TiO_2$[31,32], $W^{6+}$ for color switch for $WO_3$[33,34], and so on. Similarly, they can migrate to the above-mentioned single-atom sites for hydrogenation to replace in-situ $H_2$ dissociation.

Herein, we present mesoporous $TiO_2$ supported Pd single-atoms/nanoparticles synergistic catalyst ($Pd_{1+NPs}/TiO_2$) based on hydrogen spillover phenomenon and realize a remarkable ketone/aldehydes hydrogenation activity at room temperature. As illustrated in Fig. 1, the $Pd_{1+NPs}/TiO_2$ catalyst integrates both $Pd_1$ and $Pd_{NPs}$ sites on mesoporous $TiO_2$ supports, where the abundant $Pd_1$ sites undertake the activation task of C=O group while $Pd_{NPs}$ sites boost the dissociation of $H_2$. The H atoms generated

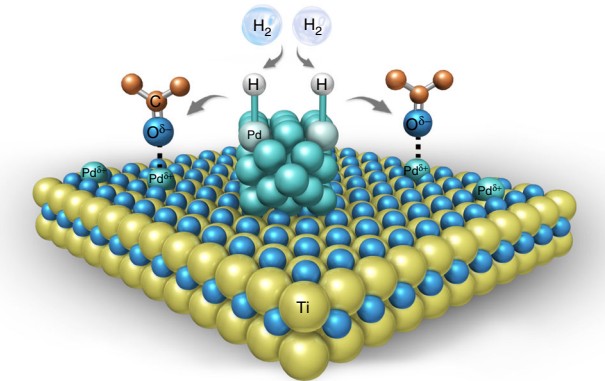

$Pd^{\delta+}$: single-atom Pd sites for C=O activation
Pd: Pd nanoparticles sites for $H_2$ dissociation

**Fig. 1 The proposed working mechanism of $Pd_{1+NPs}/TiO_2$ synergistic catalyst.** The $Pd_{1+NPs}/TiO_2$ integrates both single-atom Pd and $Pd_{NPs}$ sites on $TiO_2$ surface, in which the single-atom Pd sites undertake the activation task of C=O group while $Pd_{NPs}$ sites boost the dissociation of $H_2$. The H atoms migrate to the C=O group bounded single-atom Pd sites to complete hydrogenation. The kelly balls represent Ti atoms, the blue balls represent O atoms, the brown balls represent C atoms, the cyan balls represent the Pd atoms, and the gray balls represent the H atoms.

at $Pd_{NPs}$ sites migrate to the C=O group bounded $Pd_1$ sites to complete hydrogenation. Confirmed by the ketone/aldehydes hydrogenation at room temperature, the $Pd_{1+NPs}/TiO_2$ catalyst possesses more than twice reactivity of both simple $Pd_1/TiO_2$ and $Pd_{NPs}/TiO_2$ catalysts, and exhibits 3.2 times activity of commercial Pd/C benchmark catalyst. Moreover, the $Pd_{1+NPs}/TiO_2$ synergistic catalyst is general to various ketone/aldehydes substrates with high activity and selectivity.

## Results

**Characterizations and catalytic activity of $Pd_{1+NPs}/TiO_2$.** The $Pd_{1+NPs}/TiO_2$ synergistic catalyst was synthesized by a simple and large-scaled spray-assisted method with some modification[35,36]. The $TiO_2$ support presents anatase phase and the morphology is mesoporous microsphere with sizes of 0.4–1.0 μm (Supplementary Fig. 1). Based on the $N_2$ sorption isotherm (Supplementary Fig. 2a), the $Pd_{1+NPs}/TiO_2$ catalyst possesses a large surface of 212.8 m² g⁻¹ and pore volume of 0.36 cm³ g⁻¹ (Supplementary Fig. 2b), which supplies enough surface for the location of Pd with a loading of 4.8 wt%. As shown in high-angle annular darkfield scanning transmission electron microscopy (HAADF-STEM) image (left-up in Fig. 2a, Supplementary Fig. 3a, b), clear Pd nanoparticles (brighter spots) disperse in the mesoporous $TiO_2$ matrix. While, the energy dispersive X-ray spectroscopy (EDX)-mapping (Fig. 2a) of O (red), Ti (blue), and Pd (green) elements presents a uniform dispersion of Pd, suggesting there are atomically dispersed Pd sites. Demonstrated by the aberration-corrected HAADF-STEM (AC-HAADF-STEM) images (Fig. 2b, Supplementary Fig. 3c–e), clear single-atom Pd sites ($Pd_1$, red circle highlighted) are co-loaded nearby the nanosized Pd nanoparticles ($Pd_{NPs}$, magenta square highlighted). In addition, a few Pd rafts (yellow circle) are found in the sample. Furthermore, the Fourier transform X-ray absorption fine structure spectra (FT-EXAFS, Fig. 2c and Supplementary Fig. 4) prove the coexistence of Pd-Pd and Pd-O coordination in $Pd_{1+NPs}/TiO_2$ sample (green line) with reference to Pd foil (black line) and bulk PdO sample (magenta line). Herein, the reference of metallic Pd foil is used to validate the Pd NPs in the sample. The Pd-Pd coordination is the nature of metallic Pd bulk or NPs. We observed an apparent Pd-Pd coordination in $Pd_{1+NPs}/TiO_2$

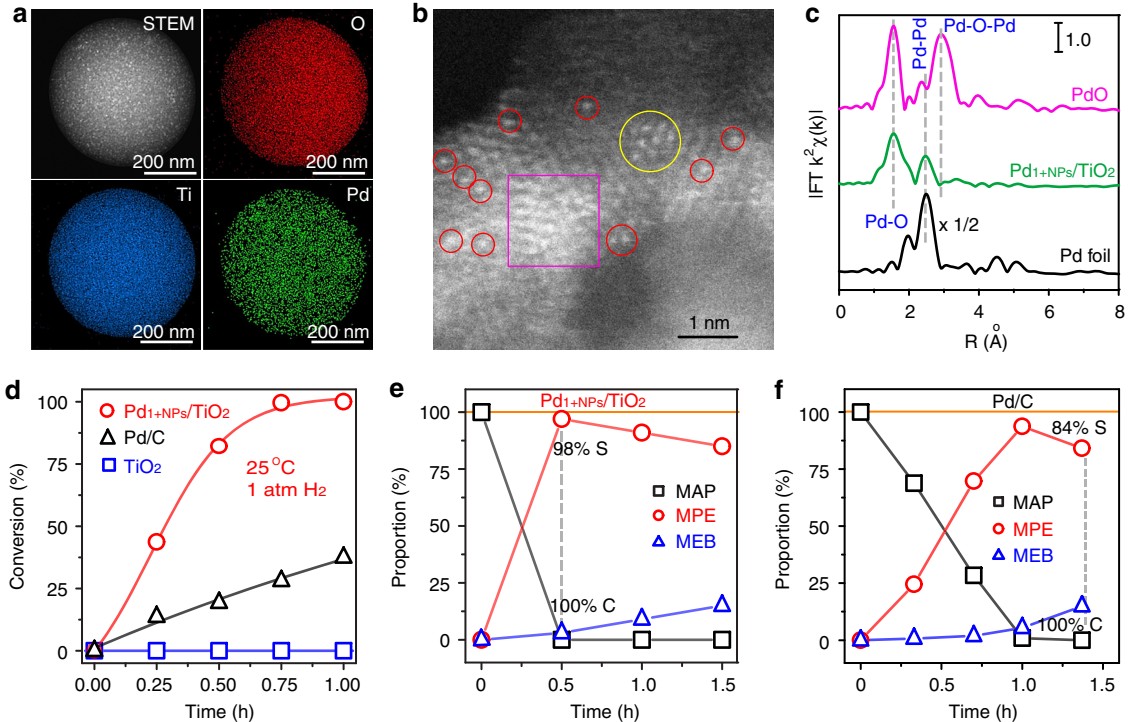

**Fig. 2 Characterizations and catalytic performance of Pd$_{1+NPs}$/TiO$_2$. a, b** HAADF-STEM-EDS mapping (**a**) and AC-HAADF-STEM image (**b**) of Pd$_{1+NPs}$/TiO$_2$. **c** R-spaced Pd K-edge FT-EXAFS spectra of Pd$_{1+NPs}$/TiO$_2$ (green line) referred to bulk Pd foil (black line) and PdO (magenta line). **d** 4-methylacetophene (MAP) conversion plots of Pd$_{1+NPs}$/TiO$_2$ (red circle), commercial Pd/C (black triangle), and TiO$_2$ catalysts (blue square) under 1 atm. H$_2$ pressure at 25 °C. **e, f** Time-dependent composition (MAP: black square; MPE (4-methylphenylethanol): red circle; MEB (4-methylethylbenzene): blue triangle) during MAP hydrogenation with Pd$_{1+NPs}$/TiO$_2$ (**e**) and Pd/C (**f**) under 0.3 MPa H$_2$ at 40 °C.

catalyst, indicating the existence of Pd NPs. Furthermore, the reference of PdO is used to confirm the contributor of Pd-O coordination. For single-atom Pd sites, only Pd-O coordination can be observed and the presence of Pd-O-Pd coordination is impossible. While, there would co-exist Pd-O and Pd-O-Pd coordination in PdO phase. In the EXAFS spectrum of Pd$_{1+NPs}$/TiO$_2$ catalyst, the Pd-O-Pd coordination disappears compared to PdO reference, suggesting the Pd-O coordination is originated from single-atom Pd sites. As shown in Supplementary Table 1 based on data fitting, the Pd-Pd coordination (2.79 Å) is assigned to the Pd$_{NP}$ species, and Pd-O coordination (2.01 Å) is assigned to Pd-O-Ti species[11,37], contributed by the Pd$_1$ sites and interfacial Pd between TiO$_2$ and Pd nanoparticles. Thus, the EXAFS study confirms the coexistence of Pd$_{NPs}$ and Pd$_1$ sites, which is well consistent with the result of AC-HAADF-STEM study. Above all, the characterizations well meet the model of Pd$_{1+NPs}$/TiO$_2$ synergistic catalyst shown in Fig. 1.

The hydrogenation performance of the Pd$_{1+NPs}$/TiO$_2$ synergistic catalyst was checked by ketone/aldehydes hydrogenation. From a case study of 4-methylacetophenone (MAP) hydrogenation shown in Fig. 2d and Supplementary Table 2, the Pd$_{1+NPs}$/TiO$_2$ catalyst (red plots) exhibits a remarkable activity at 25 °C under 1 atm. H$_2$ pressure. The substrate gets 100% converted within 1.0 h with a high alcohol selectivity of 99%. As a reference, the activity of pristine mesoporous TiO$_2$ (Supplementary Fig. 5) is negligible (blue plots). To position the Pd$_{1+NPs}$/TiO$_2$ in terms of ketone/aldehydes hydrogenation performance, commercial Pd/C (5.2 wt%, black plots) benchmark catalyst (Supplementary Fig. 6) was studied. We can see that the reactivity of Pd$_{1+NPs}$/TiO$_2$ is 3.2 times better. The turnover frequency (TOF, based on the conversion at 15 min normalized by Pd dispersion)[38] value reaches 4362 h$^{-1}$, one order of magnitude higher than the

reported results[14,15,39,40]. The 4-methylphenylethanol (MPE) and 4-methylethylbenzene (MEB) are the detected product from MAP hydrogenation. To show the superior selectivity of Pd$_{1+NPs}$/TiO$_2$, the MAP hydrogenation reactions were performed at 40 °C under 0.3 MPa H$_2$. As for Pd$_{1+NPs}$/TiO$_2$ (Fig. 2e), the MAP conversion reaches 100% within 20 min and the alcohol (MPE) selectivity keeps a high level of 98%, which is much better than Pd/C benchmark (Fig. 2f) with 84% of MPE selectivity. The production of MEB mainly appears after complete formation of alcohol. While for Pd/C, the MEB formation appears synchronously with the production of alcohol. Therefore, the proposed Pd$_{1+NPs}$/TiO$_2$ synergistic catalyst achieves great success in active and selective hydrogenation of ketone/aldehydes to alcohol under mild conditions.

**Confirming the synergistic function of Pt$_1$ and Pt$_{NPs}$.** To verify the role of the Pd$_1$ and Pd$_{NPs}$ in the hydrogenation behaviors, we designed additional two model catalysts: atomically Pd$_1$/TiO$_2$ (Fig. 3a) and nanosized Pd$_{NPs}$/TiO$_2$ (Fig. 3b). With reference to Pd$_{1+NPs}$/TiO$_2$ (Fig. 3c), Pd$_1$/TiO$_2$ has no Pd nanoparticles and Pd$_{NPs}$/TiO$_2$ has no single-atom Pd. The sample of Pd$_1$/TiO$_2$ (Supplementary Fig. 7) was prepared by same method to Pd$_{1+NPs}$/TiO$_2$ except lowing the Pd loading to 1.0 wt%. The Pd$_{NPs}$/TiO$_2$ (Supplementary Fig. 8) was obtained by pre-reducing Pd(NO$_3$)$_2$ to Pd NPs with H$_2$ before spray-pyrolysis. The Pd states were studied by the spectra of X-ray absorption near-edge structure (XANES, Fig. 3d). With references of Pd foil (black line) and bulk PdO (orange line), the intensity of while line indicates the Pd$_1$/TiO$_2$ is the most positive-charged, and subsequently Pd$_{1+NPs}$/TiO$_2$ (magenta line) and Pd$_{NPs}$/TiO$_2$ (blue line), reflecting the relative content of metallic Pd and Pd-O-Ti species[41]. The

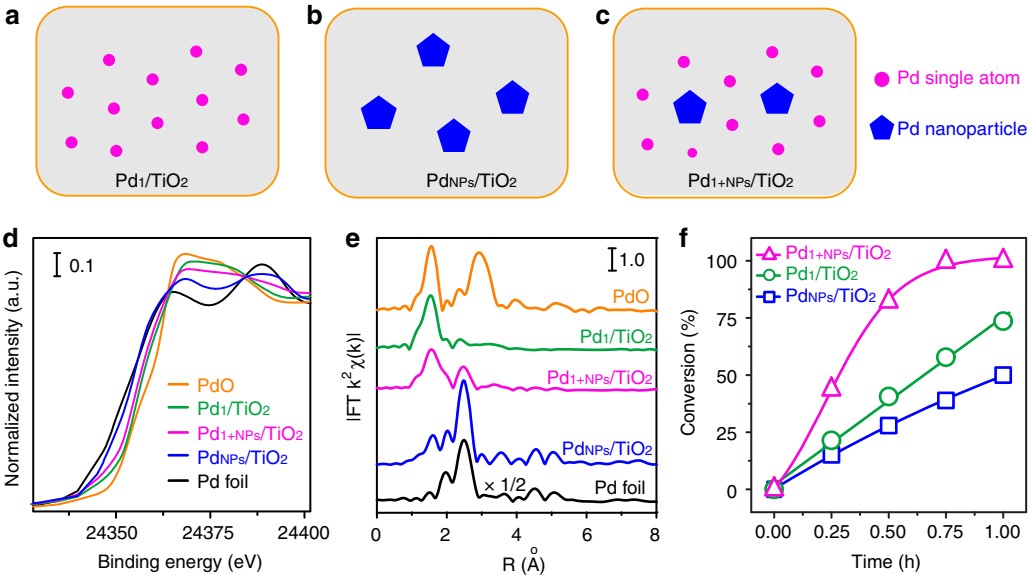

**Fig. 3 Structures and performance of differently dispersed Pd/TiO₂ catalysts. a–c** Illustration of $Pd_1/TiO_2$ (**a**), $Pd_{NPs}/TiO_2$ (**b**), and $Pd_{1+NPs}/TiO_2$ (**c**) catalysts, where Pd single atom and Pd nanoparticle are represented by solid magenta circle and blue pentagon. **d, e** Pd foil (black line) and bulk PdO (orange line) referenced XANES (**d**) and R-spaced FT-EXAFS (**e**) spectra of $Pd_1/TiO_2$ (green line), $Pd_{NPs}/TiO_2$ (blue line) and $Pd_{1+NPs}/TiO_2$ (magenta line). **f** MAP hydrogenation reaction plots of $Pd_1/TiO_2$ (green circle), $Pd_{NPs}/TiO_2$ (blue square), and $Pd_{1+NPs}/TiO_2$ (magenta triangle) at 25 °C under 1 atm. H₂.

EXAFS study (Supplementary Fig. 9) and fitting (Supplementary Fig. 10 and Supplementary Table 1) further clear the coordination of Pd. As shown in R-spaced FT-EXAFS spectra (Fig. 3e) with reference to Pd foil (black line) and bulk PdO (orange line), $Pd_{NPs}/TiO_2$ (blue line) possesses much more metallic Pd-Pd coordination (2.83 Å) than that in $Pd_{1+NPs}/TiO_2$ (magenta line), agreeing with the full nanosized dispersion of Pd. The weak Pd-O coordination (2.00 Å) is contributed by the interfacial Pd-O-Ti. Moreover, the Pd in $Pd_1/TiO_2$ (green line) nearly presents total Pd-O coordination (2.01 Å) without neither Pd-Pd nor Pd-O-Pd coordination, confirming that $Pd_{1+NPs}/TiO_2$ indeed possesses atomic dispersion of Pd which is bonded to the surface oxygen of TiO₂ support[42,43].

The three model catalysts with $Pd_1$ and $Pd_{NPs}$ gradient serve as the platform to study the response of $Pd_1$ and Pd-Pd sites to hydrogenation activities. To make the comparison meaningful, the ratio of Pd to the substrate (0.2 mol%) was controlled equally, and the reactions were performed under same conditions (25 °C, 1 atm. H₂). Typically, 5 mg and 21 mg of $Pd_{1+NPs}/TiO_2$ and $Pd_1/$ TiO₂ were used for one pot reaction with 1 mmol of MAP substrate, respectively. As shown in Fig. 3f, the absence of the $Pd_1$ site of $Pd_{NPs}/TiO_2$ (blue line) causes 67% drop of reactivity compared to $Pd_{1+NPs}/TiO_2$ (magenta line), revealing that atomically dispersed $Pd_1$ site is crucial to get high performance of ketone/aldehydes hydrogenation. Furthermore, the $Pd_1/TiO_2$ (green line) with few Pd NPs presents 51% drop of reactivity, suggesting that Pd NPs is essential to ketone/aldehydes hydrogenation as well. By the way, the zero kinetics of $Pd_1/TiO_2$ is well consistent to the observation of Zheng's group[11]. In addition, to exclude the $Pd_1$ aggregation induced the low activity of $Pd_1/TiO_2$, the used $Pd_1/TiO_2$ catalysts were further studied by AC-HAADF-STEM images (Supplementary Fig. 11). Compared with fresh $Pd_1/TiO_2$ SAC sample (Supplementary Fig. 7), only a few Pd single-atoms aggregate to rafts or clusters (~1 nm) while no Pd nanoparticle appear in the used sample, indicating that the stability of Pd single atoms are acceptable to discuss their activity under our hydrogenation conditions. Therefore, the above results strongly and clearly point to a synergistic function of the $Pd_1$ and

$Pd_{NPs}$ in $Pd_{1+NPs}/TiO_2$ catalysts toward ketone/aldehydes hydrogenation.

To further clear the role of $Pd_1$ site and Pd NPs, their TOF values based on active Pd sites were evaluated. As shown in Supplementary Table 2, the dispersions of $Pd_{1+NPs}/TiO_2$ and $Pd_{NPs}/TiO_2$ catalysts were measured as 17.7% and 6.2%, respectively. As for the $Pd_1/TiO_2$, we find the CO pulse chemisorption is inapplicable to determine the dispersion of Pd because the weak binding of CO on single-atom dispersed metal leads to large desorption of CO[44,45]. We estimated the Pd dispersion as 67% for 1% $Pd_1/TiO_2$ by leaching the surface Pd with a mixture of H₂O₂ and HCl solution, which is well consistent with the result in Pratsinis et al.' work[46]. As summarized in Supplementary Table 2, the $Pd_1/TiO_2$ exhibited a lowest activity with TOF value of 645 h⁻¹. Moreover, we found that the TOF value of $Pd_{NPs}/TiO_2$ (4565 h⁻¹) were highly close to that of $Pd_{1+NPs}/TiO_2$ (4361 h⁻¹) catalyst, which means the high activity of ketone/aldehydes hydrogenation on Pd NPs. However, limited by the nature of low dispersion of NPs, $Pd_{NPs}/TiO_2$ presents low-level reactivity. On the other side, although single-atom catalysts possess much higher metal efficiency, the reactivity of $Pd_{NPs}/TiO_2$ also presents low-scale due to the low activity of $Pd_1$ sites. The synergistic function of large numbered $Pd_1$ sites and partial $Pd_{NPs}$ additive brings a remarkable performance for ketone/aldehydes hydrogenation.

**Understanding the synergistic function of $Pt_1$ and $Pt_{NPs}$.** To understand the synergistic function of the $Pd_1$ and $Pd_{NPs}$ in $Pd_{1+NPs}/TiO_2$ catalysts, we studied the key steps of C=O adsorption and H₂ dissociation by the density functional theory (DFT) calculation based on two models of $Pd_1/TiO_2$ (110) and metallic Pd (111) surfaces. According to the X-ray diffraction of $Pd_{1+NPs}/TiO_2$ catalysts (Supplementary Fig. 1c), the anatase TiO₂ and face-centered cubic Pd was chosen to build the model for TiO₂ (110) and Pd (111) surfaces, respectively. Herein, the van der Waals (vdW) interactions with Grimme method (B97-D) for DFT-D correction were considered for all the calculations[47]. To produce a more precise calculation than GGA, the

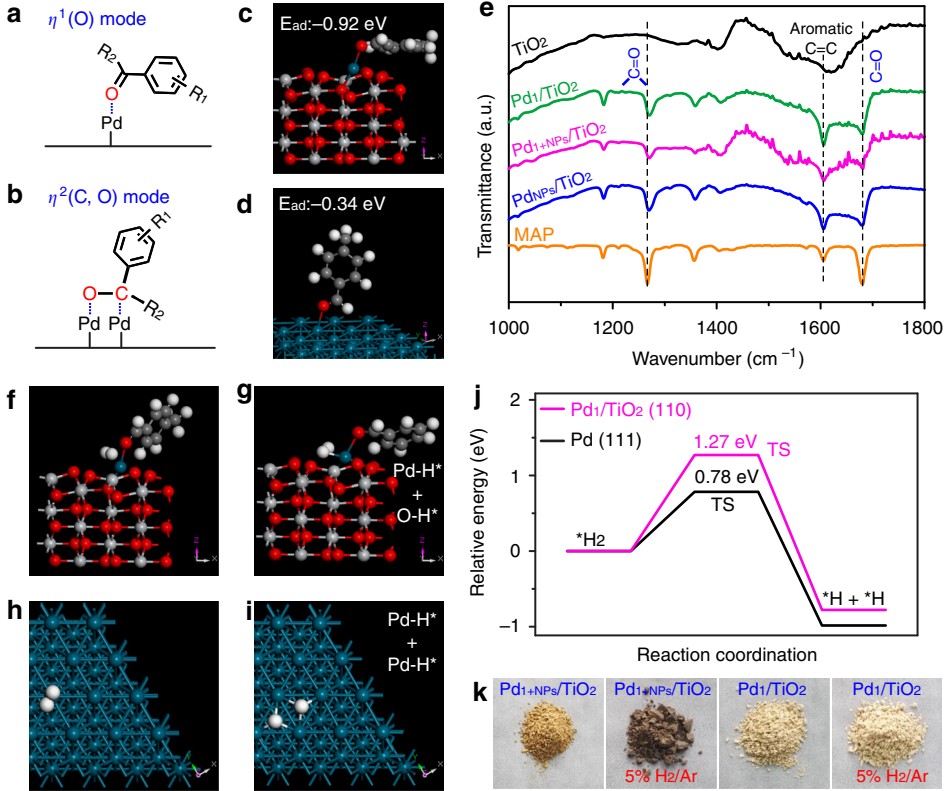

**Fig. 4 Mechanism study of Pd$_{1+NPs}$/TiO$_2$ for ketone/aldehydes hydrogenation. a, b** Illustration of $\eta^1$ (O) (**a**) and $\eta^2$ (C, O) (**b**) configuration of C=O adsorption. **c, d** Models of C=O adsorbed on Pd$_1$/TiO$_2$ (110) (**c**) and Pd (111) (**d**) surface. **e** ATR-FTIR spectra of MAP substrate (orange line) and MAP adsorbed on TiO$_2$ (black line), Pd$_1$/TiO$_2$ (green line), Pd$_{1+NPs}$/TiO$_2$ (magenta line), and Pd$_{NPs}$/TiO$_2$ (blue line) catalysts. **f–i** Adsorption models of H$_2$ molecule (**f, h**) and H atoms (**g, i**) on substrate covered Pd$_1$/TiO$_2$ (110) (**f, g**) and Pd (111) (**h, i**) surfaces. **j** Relative energy plots of H$_2$ dissociation on Pd$_1$/ TiO$_2$ (110) (magenta line) and Pd (111) (black line) surfaces. **k** Digital photographs of Pd$_{1+NPs}$/TiO$_2$ and Pd$_1$/TiO$_2$ powder before and after 5% H$_2$/Ar flow for 2 min at room temperature.

GGA + U approach was performed to deal with all the TiO$_2$-relative calculations, in which an intra-atomic electron–electron interaction was introduced as an on-site correction to describe systems with localized d and f states[48]. According to Li et al.'s work[49], a U value of 7 eV for Ti atoms was considered. For C=O adsorption, two adsorption modes, $\eta^1$ (O) (Fig. 4a) and $\eta^2$ (C, O) (Fig. 4b) are possible[14]. The DFT results show $\eta^1$ (O) mode is chosen on both Pd$_1$/TiO$_2$ (Fig. 4c) and Pd NPs (Fig. 4d). We can find the C=O adsorption energy (E$_{ad}$) on Pd$_1$/ TiO$_2$ (−0.92 eV) is much higher on metallic Pd surface (−0.34 eV), suggesting more efficiency of CO adsorption on Pd$_1$ site. Besides, we studied the adsorption of MAP molecule on Pd$_{1+NPs}$/TiO$_2$ surface by ATR-FTIR (attenuated total internal reflectance Fourier transform infrared spectroscopy). As shown in Fig. 4e, with reference to pure MAP molecule (orange curve), the presence of the stretch and bend vibration of C=O group centered at 1679 and 1265 cm$^{-1}$ indicates $\eta^1$ (O) mode is preferentially formed because the C=O adsorption would be canceled if $\eta^2$ (C, O) configuration presents[50]. In addition, a blue-shift of 3 cm$^{-1}$ for C=O bend vibration reveals the strong adsorption, where a rigid structure of C=O-Pd causes enhanced energy for C=O bend vibration. For meaningful comparison, the C=O adsorption on Pd$_1$/TiO$_2$ (green curve), Pd$_{NPs}$/TiO$_2$ (blue curve) and TiO$_2$ (black curve) were also studied. The MAP adsorption on Pd$_1$/TiO$_2$ and Pd$_{NPs}$/TiO$_2$ is similar to Pd$_{1+NPs}$/TiO$_2$, where we can also observe the C=O bend vibrational shift. However, the MAP adsorption TiO$_2$ were too weak to be detected by our facility.

The DFT study of the other essential step of H$_2$ dissociation is shown in Fig. 4f–j. The H$_2$ dissociation is an uphill process, and the activation energies (E$_a$) are calculated for the models of C=O group bonded Pd$_1$/TiO$_2$ (110) (Fig. 4f) and metallic Pd (111) (Fig. 4h) surface. It is interesting that H$_2$ dissociation on Pd$_1$/TiO$_2$ goes heterolytic pathway (Fig. 4g) while it presents homolytic dissociation (Fig. 4i) on metallic Pd (111) surface. The heterolytic pathway on single-atom sites has been found by Zheng's group[11], but we further find that the barrier of H$_2$ dissociation (Fig. 4j) gets much increased on Pd$_1$/TiO$_2$ (1.27 eV) compared to metallic Pd surface (0.78 eV). This suggests that H$_2$ dissociation is difficult at Pd$_1$ site and primarily goes on Pd NPs. Moreover, the difference of their interaction with H$_2$ is visible. Figure 4k shows the digital photographs (k) of Pd$_{1+NPs}$/TiO$_2$ and Pd$_1$/TiO$_2$ powder before and after 5% H$_2$/Ar flow for two minutes at room temperature. Vividly, the single-atom Pd is hardly to be reduced at room temperature, just because of the high dissociation barriers of H$_2$ molecule to H atoms. While, the Pd$_{1+NPs}$/TiO$_2$ presents active interaction with H$_2$ owning to the presence of Pd nanoparticles.

According to DFT results, H$_2$ will dissociate preferentially on metallic Pd nanoparticles and ketone/aldehydes molecules can be adsorbed at Pd$_1$ sites. In addition to the TOF values of Pd$_1$/TiO$_2$, Pd$_{NPs}$/TiO$_2$, and Pd$_{1+NPs}$/TiO$_2$ catalysts, the H$_2$ dissociation is more important than C=O group activation because the TOF values of Pd$_{NPs}$/TiO$_2$ and Pd$_{1+NPs}$/TiO$_2$ are approximately equal although their C=O adsorption energies are quite different. To clear the synergy function of Pd$_1$ atoms and NPs more clearly, we clarified it with a tandem working mechanism of Pd$_1$ atoms and

**Table 1 The performance of $Pd_{1+NPs}/TiO_2$ synergistic catalyst toward different substrates.**

| Entry | Substrate | Product | Conversion[a] % (reaction time) | Selectivity[b] % | TOF h⁻¹ |
|---|---|---|---|---|---|
| 1. | | | 100 (1 h) | 98 | 4362 |
| 2. | | | 100 (1 h) | 99 | 4146 |
| 3. | | | 100 (1 h) | >99.7 | 4610 |
| 4. | | | 100 (1 h) | >99.5 | 4436 |
| 5. | | | 100 (0.25 h) | 98 | 16288 |
| 6. | | | 100 (0.5 h) | >99.5 | 5115 |
| 7. | | | 98.2 (0.75 h) | 74 | 4553 |
| 8. | | | 100 (1 h) | 72 | 3035 |

[a]The catalytic evaluation was performed with 1 mmol substrate, 5 mL ethanol and 5 mg Pd/TiO₂ catalyst with Pd loading of 4.8 wt% under 1 atm. H₂ pressure at 25 °C
[b]The selectivity was evaluated at the conversion of 100%

NPs. As shown in Fig. 1, the $H_2$ molecules are dissociated into H atoms on Pd NPs with fast kinetics. The spillover of H atoms from Pd NPs migrates to the ketone/aldehydes-adsorbed $Pd_1$ sites. The hydrogenation was finally completed at $Pd_1$ sites. Herein, both Pd NPs and $Pd_1$ atoms are key to high hydrogenation performance. In terms of the $Pd_{NPs}/TiO_2$ with high activity, the nature of low dispersion caused a low hydrogenation exhibition. While, although atomically $Pd_1/TiO_2$ catalyst possessed the highest dispersion, the high $H_2$ dissociate barrier induced low activity still made it present a low-level hydrogenation performance. The tandem working mechanism dissolved the disadvantages of both single-atom $Pd_1$ site (low activity) and Pd NPs (low dispersion) synergistically. As observed above, the $Pd_{1+NPs}/TiO_2$ synergistic catalyst present more than twice reactivity of simple $Pd_1/TiO_2$ and $Pd_{NPs}/TiO_2$ catalysts. According to Crook et al.'s work[51,52] and the working mechanism of $Pd_1$ atoms and NPs, the size of Pd NPs and ratio of Pd NPs/$Pd_1$ sites would affect the hydrogenation performance of $Pd_{1+NPs}/TiO_2$ catalyst.

**The universality and stability of $Pd_{1+NPs}/TiO_2$ catalysts.** The high activity is also confirmed by the hydrogenation of other ketone/aldehydes (Table 1 and Supplementary Figs. 12–20). Acetophenone was observed to give the corresponding α-phenethyl alcohol in quantitative yield like MAP (Table 1, entries 1 and 2). The 3-methylacetophene was reduced to the corresponding alcohol in 99.7% yield (Table 1, entry 3). 1-phenyl-1-propanone could also be reduced to 1-phenyl-1-propanol in 99.5% yield (Table 1, entry 4). Benzaldehyde could also be reduced at mild condition with almost completer conversion (Table 1, entries 5 and 6), and even an unusual TOF value of

16,288 h⁻¹ is acquired for benzaldehyde. Furfuryl alcohol is an important fine chemical, which is widely used in synthetic fibers, rubber and pesticides industries[53]. Using $Pd_{1+NPs}/TiO_2$ synergistic catalyst, furfural could be reduced into furfuryl alcohol in 98.2% conversion and more than 74% selectivity under 1 atm. $H_2$ pressure at 25 °C (Table 1, entry 7). Cinnamyl alcohol was obtained in high conversion but moderate selectivity (Table 1, entry 8). The unique adsorption and activation mode of the catalyst for carbonyl and hydrogen, resulted in the universality of $Pd_{1+NPs}/TiO_2$ synergistic catalyst for ketone/aldehydes hydrogenation at room temperature and atmospheric pressure.

The stability of $Pd_{1+NPs}/TiO_2$ catalysts were evaluated by measuring the conversion at 0.5 h at each cycle of reusing. At each cycle, pure substrate was added into the reaction cell after completing the last reaction. As shown in Supplementary Fig. 21, the reactivity drops gradually with the increase of cycle. At fifth cycle, 45% of the reactivity was lost in comparison with the first cycle. However, we should caution that the reactivity of catalysts was also affected by the accumulated product induced chemical equilibrium because the product was also not separated out during the five cycles. To further evaluate the limit of $Pd_{1+NPs}/TiO_2$ catalysts, the substrate concentration was increased to 5 mmol without changing any other conditions, so the ratio of Pd to substrate was correspondingly decreased to 0.04 mol%. As shown in Supplementary Fig. 22, the reactivity of $Pd_{1+NPs}/TiO_2$ still kept high level although five times of substrate were added. The alcohol selectivity also reached more than 98%. However, resulted from the insufficient $H_2$ diffusion in solvent, the TOF value (4084 h⁻¹) presented slightly drop in comparison with that with 1 mmol substrate (4362 h⁻¹). Above all, the $Pd_{1+NPs}/TiO_2$ catalyst exhibit acceptable stability to act as a reusable catalyst for ketone/aldehydes hydrogenation.

After the synergistic function of $Pd_1$ sites and Pd NPs, we found that the $TiO_2$ support played another key role in the ketone/aldehydes hydrogenation. As shown in Supplementary Fig. 23, the activity was negligible when Pd was supported on $Al_2O_3$, NiO, $Fe_2O_3$, $Mn_3O_4$, etc. Whatever the dispersion of Pd, it is an interesting result in comparison with anyone of $Pd_1/TiO_2$, $Pd_{NPs}/TiO_2$, and $Pd_{1+NPs}/TiO_2$ catalysts. The systematically study on the fundamental knowledge of support-dependent activity would be presented in the future work.

In summary, this work proposes and confirms a highly efficient $Pd_{1+NPs}/TiO_2$ synergistic catalyst for selective ketone/aldehydes hydrogenation at room temperature and ambient $H_2$ pressure. The high activity is contributed by the synergistic function of $Pd_1$ and $Pd_{NPs}$ on $TiO_2$ that the abundant $Pd_1$ sites undertake the activation task of C=O group while $Pd_{NPs}$ sites boost the dissociation of $H_2$. This work not only paves the way to green production of alcohols from selective ketone/aldehydes hydrogenation with $Pd_{1+NPs}/TiO_2$ catalyst, but also enlighten people in the fundamental knowledge of ketone/aldehydes hydrogenation on variously dispersed Pd/TiO₂ catalysts.

## Methods

**Materials preparation.** For $Pd_{1+NPs}/TiO_2$ catalyst, 1.0 mL concentrated hydrochloric acid (HCl, 35–37%), 0.5 g F127 (Sigma-Aldrich), 6.60 mL of $Pd(NO_3)_2$ (30 mM, Aladdin), and 1.36 g (4 mmol) tetrabutyl titanate (TBOT, Aladdin) are added into 34 mL deionized water in order. A mixture solution was obtained with ultrasonic assistance. The mixture was then sprayed into a tube furnace (600 °C). The powders were collected by a filter paper collector by a pump, and finally calcined at 400 °C for 2 h in air with heating rate of 2 °C/min. The final catalysts were obtained after washing the calcined powders with deionized water and absolute ethanol.

The preparation of $Pd_1/TiO_2$ catalyst is same to that of $Pd_{1+NPs}/TiO_2$ catalyst except the amount of $Pd(NO_3)_2$ (30 mM) is changed to 1.32 mL.

The preparation of $Pd_{NPs}/TiO_2$ catalyst is same to that of $Pd_{1+NPs}/TiO_2$ with some modification. In detail, 1.0 mL hydrochloric acid (35–37%), 0.5 g F127, 6.60 mL of $Pd(NO_3)_2$ (30 mM), and 1.36 g TBOT are added into 34 mL deionized water

in order. The $Pd(NO_3)_2$ in the mixture was pre-reduced to Pd nanoparticles by continuous flow of $H_2$ (99.999%) for 5 min at room temperature. The follow processes are same to that of $Pd_{1+NPs}/TiO_2$.

The preparation of mesoporous $TiO_2$ catalyst is the same as $Pd_{1+NPs}/TiO_2$ catalysts without addition of $Pd(NO_3)_2$ solution.

The final Pd loadings in the catalysts were determined by the inductively coupled plasma-atomic emission spectrometry (ICP-AES, Supplementary Fig. 24).

All the reagents were used as received without further purification. The resistivity of used water was more than 18.25 MΩ·cm.

**Materials characterizations**. All the catalysts were activated before characterizations and catalysis tests. The activation was performed at room temperature by 5% $H_2/Ar$ mixture for 3–5 min. During the activation, the color of the powder changes from light-yellow to dark-brown.

The materials are characterized with by SEM (Hitachi, S-4800, Japan) with 5 kV accelerate voltage and TEM (Technai G20 S-TWIN, FEI) with 200 kV accelerate voltage, high-angle annular darkfield scanning transmission electron microscopy (HAADF-STEM, JEOL2100), X-ray powder diffraction (XRD, D8, Brook) with Cu $K_\alpha$ radiation ($\lambda = 0.15418$ nm) operated at 40 kV and 40 mA, and HAADF-STEM EDS-mapping. The aberration-corrected high-angle annular darkfield scanning transmission electron microscopy (AC-HAADF–STEM) images were performed on JEM-ARM200F (JEOL) with accelerating voltage of 200 kV. The $N_2$ sorption isotherms (77 K) with TriStar II 3020 V1.03 (Micromeritics, USA) was performed to analyze the Brunauer–Emmett–Teller (BET) specific surface area and Barrett–Joyner–Halenda (BJH) pore size distributions.

**Pd dispersion measurement**. The dispersion of Pd was determined by CO pulse chemisorption. Firstly, catalysts were pre-treated in He flow at 200 °C for 1 h. Subsequently, the catalysts were further reduced at 50 °C with 10% $H_2/Ar$ for 1 h. The cell was cleaned with He flow to remove $H_2/Ar$. Finally, the CO pulse adsorption was performed at 50 °C and stopped the pulse until the no CO adsorption appeared on the samples. The accumulate CO adsorption was used for the calculation of dispersion. The Pd loading for calculating the dispersion was measured by ICP-AES with the samples prepared in the same batch to Pd dispersion measurement.

To estimate the Pd dispersion in 1% $Pd/TiO_2$ by leaching the surface Pd, 16.5 mg of catalyst powder was dispersed into 5 mL of mixture solution containing 6% $H_2O_2$ and 7% HCl. The solution was treated at 60 °C for 2 h and constantly volumed to 100 mL. The solution was separated by a filter to remove the catalyst solid. The leached Pd was subsequently measured by ICP-AES. The Pd dispersion was defined as (leached Pd in solution)/(total Pd in catalyst).

The number of active sites were calculated based on the amount of Pd usage for one pot hydrogenation reaction (0.002 mmol) and Pd dispersion.

**Catalytic hydrogenation tests**. 1 mmol substrate (AR, Aladdin) was dissolved in 5 mL ethanol (99.8%) in 25 mL glassy cell, and 5 mg 4.8 wt% $Pd_{1+NPs}/TiO_2$ was dispersed by ultrasonic assistance. The gas in the cell was replaced by high-purity $H_2$ (99.999%) five times under a pressure of 0.3 MPa. For the hydrogenation reaction, an ambient $H_2$ pressure is performed with a flow of about 10 mL/min controlled by a mass flowmeter (MFC). The reaction temperature was controlled by a cycling water. The magnetic stirring speed was 1000 rpm to reduce the diffusion effect. The conversion and selectivity were qualitatively determined by a gas chromatography (GC) equipped with a hydrogen flame ionization detector (FID). The well linear relationship between FID responded peak area and analyte concentration was shown in Supplementary Fig. 25.

The amount of metal Pd is kept the same to that in 5 mg of 4.8 wt% $Pd_{1+NPs}/TiO_2$ for the evaluation of other Pd-based catalysts. The ratio of Pd to substrate was controlled as 0.2 mol%.

The stability of $Pd_{1+NPs}/TiO_2$ catalysts were evaluated by measuring the conversion at 0.5 h for five cycles. At each cycle, 1 mmol of pure substrate was added into the reaction cell after completing the last reaction.

**ATR-FTIR study**. Typically, 0.1 g of catalyst powders were immersed in 10 mL of 0.2 mol/L 4-methylacetophenone (solvent: ethanol) for 12 h to acquire the adsorption equilibrium. The catalyst was separated by centrifugation, washed with absolute ethanol, and dried at 60 °C in vacuum box for 6 h. The powders were pressed to form a platelet. The attenuated total internal reflectance Fourier transform infrared spectroscopy (ATR-FTIR) were collected on a Thermo Scientific™ Nicolet iS5 FTIR Spectrometer with a resolution of 0.5 cm$^{-1}$ at room temperature.

**X-ray absorption spectra collection and data processing**. The extended X-ray absorption fine structure (EXAFS) measurements at the Palladium K-edge were measured in transmission mode at the beamline BL14W1 station of SSFR (Shanghai Synchrotron Radiation Facility), P. R. China. All samples were pelletized as disks of 13 mm diameter with 1 mm thickness using graphite powder as a binder.

The acquired EXAFS data were processed according to the standard procedures using the ATHENA module implemented in the IFEFFIT software packages[54]. The EXAFS spectra were obtained by subtracting the post-edge background from the overall absorption and then normalizing with respect to the edge jump step. Subsequently, the $\chi(k)$ data were Fourier transformed to real ($R$) space using a hanning windows ($dk = 1.0$ Å$^{-1}$) to separate the EXAFS contributions from different coordination shells. The quantitative information can be obtained by the least-squares curve fitting in the $R$ space with a Fourier transform $k$ space range of 2.4–13.2 Å$^{-1}$, using the module ARTEMIS of programs of IFEFFIT. The backscattering amplitude F($k$) and phase shift $\Phi(k)$ were calculated using FEFF8.0 code.

**Method and model for DFT calculation**. The first principles calculations in the framework of density functional theory, including structural, electronic performances, were carried out based on the Cambridge Sequential Total Energy Package known as CASTEP[55]. The exchange–correlation functional under the generalized gradient approximation (GGA)[56] with norm-conserving pseudopotentials and Perdew–Burke–Ernzerhof (PBE) functional was adopted to describe the electron–electron interaction[57]. The van der Waals (vdW) interactions with Grimme method (B97-D) for DFT-D correction were considered for all the calculations[47]. To produce a more precise calculation than GGA, the GGA + U approach was performed to deal with all the $TiO_2$-relative calculations, in which an intra-atomic electron–electron interaction was introduced as an on-site correction to describe systems with localized d and f states[48]. According to Li et al.'s work[49], a U value of 7 eV for Ti atoms was considered. An energy cutoff of 750 eV was used and a $k$-point sampling set of $5 \times 5 \times 1$ were tested to be converged. A force tolerance of 0.01 eV Å$^{-1}$, energy tolerance of $5.0 \times 10^{-7}$ eV per atom and maximum displacement of $5.0 \times 10^{-4}$ Å were considered. The vacuum space along the $z$ direction is set to be 15 Å, which is enough to avoid interaction between the two neighboring images. The bottom three atomic layers were fixed, and the top three atomic layers were relaxed. According to the XRD pattern (Supplementary Fig. 1c), anatase $TiO_2$ crystal was chosen to cleave the stable $TiO_2(110)$ surface and face-centered cubic (FCC) Pd crystal was used to cleave Pd(111) surface. The model of $(3 \times 2)$ supercell slab ($Ti_{36}O_{72}$) including six atomic layers was adopted for anatase $TiO_2(110)$ surface. The termination of $TiO_2(110)$ surface contains the 4-fold coordinated Ti ($Ti_{4c}$) and 2-fold coordinated O ($O_{2c}$). The model of $(4 \times 4)$ supercell slab ($Pd_{96}$) including six atomic layers was adopted for FCC Pd(110) surface. Then, the $C_8H_8O$ and $H_2$ had been absorbed on the surface of Pd(111) and $Pd_1/TiO_2(110)$. Besides, the complete LST/QST search protocol and the SCF tolerance of $5.0 \times 10^{-7}$ eV per atom are set for transition states.

Adsorption energy $\Delta E$ of A group on the surface of substrates was defined as

$$\Delta E = E_{*A} - (E_* + E_A) \tag{1}$$

where $*A$ and $*$ denote the adsorption of A group on substrates and the bare substrates, $E_A$ denotes the energy of A group. The obtained adsorption energies were listed in Supplementary Table 3.

## Data availability

The additional data are provided in the Supplementary Information. All the data that support the findings of this study are available from the corresponding author upon reasonable request.

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

## Acknowledgements

This work was supported by the National Natural Science Foundation of China (21801003, 21871005, 21471006), the Research Start-up and Outstanding Youth Foundation of Anhui Polytechnic University (2016YQQ019, 2019JQ01), Natural Science Foundation of Anhui Province (1808085QB47), Natural Science Foundation of Anhui Provincial Education Department (KJ2017A111, KJ2017A112). The authors thank the National Synchrotron Radiation Laboratory (Shanghai, P. R. China). Authors thank Mr. Ming Zuo in University of Science and Technology of China (USTC) for the professional support of AC-HAADF-STEM observations and Hangzhou Precision New Materials and Technology Co., Ltd (Hangzhou, China) for the DFT study.

## Author contributions

B.G. and Y.L. conceived this work. L.K., Z.C., and S.L. collected the data. E.K., N.Y., Y.R., C.F., and X.L. analysis the data. L.K. and Z.C. wrote the paper, and all the authors contributed the paper revisions. L.K. and Z.C. contributed equally to this work.

## Competing interests

The authors declare no competing interests.
