## [Peer Review File · Nature Communications]

Reviewer #1 (Remarks to the Author):

The manuscript from Li et al., describes the development of synergistic catalysts that comprise of isolated Pd sites and Pd NPs on a TiO₂ surface and they propose a mechanism where ketones or aldehydes are bound by the single Pd sites and activated towards reduction with H₂ which is bound and cleaved by the Pd NPs.

The introduction appears somewhat limited in scope and should either be broadened to include traditional heterogeneous catalysis as that is stated in the abstract as one of the goals – to replace Pd/C. In addition, the reference material should be revised to substantiate and validate the claims. Ref 3 covers MOFs for selectivity of C=O reduction over C=C bonds and photochemical production of Pd NPs. Ref 4 concerns the number of isomers produced on reduction of hexanone which I don't necessarily consider by-products, just low selectivity.

The characterization of the catalysts appears thorough but when looking at the HAADF-STEM image in Fig 2b, its hard for my untrained eye to definitively see that the circles are definitely single Pd atoms. I looked at work by Tao Zhang in this area (see Fig 1 in Nat. Chem., 3, 634 & in Accounts in Chemical Research, 2013, 46, 8, 1740) and they show a clear difference in the image which is lacking in this figure. Due to this resolution difference, how do the authors rationalize the single atom Pd's in circle and the Pd NPs the boxes? Again, in the work of Zhang these seem significantly different. Do the authors observe any Pd rafts?

The work to make Pd single atoms and Pd NPs independently is a good way to validate the catalyst and the EXAFS work should be described a little more comprehensively in terms of why the PdO and Pd foil were also analyzed and why those are significant and why they guided the characterization.

In Figure 3f the caption states that the amount of Pd to substrate is maintained constant – the details for this and the amounts and conditions should be explicitly provided to allow the reader to qualify the results.

Table 1 shows some very impressive results. However no supporting data is provided. I would need to see spectra and an in depth description of the analytical process to validate this work. I would also like to know if the FID has been calibrated – while more accurate than TIC knowing this is accurate would be appreciated. While the authors talk through the table extensively (this could be shortened) there is no real substance to this section and without the data its hard to say whether these results are accurate.

Have these catalysts been reused and recycled in other experiments or upon completion have the authors added more substrate to test the limits of catalyst use? This should be added to the manuscript.

Have any other surfaces or supports been investigated? If not why only TiO₂?

Thanks, Andrew Sutton, Los Alamos National Lab.

Reviewer #2 (Remarks to the Author):

In this work, the authors reported that coexistence of Pd1 single atoms and NPs on TiO₂ support would induce the synergistic function for hydrogenation of ketone/aldehydes up to two times of activity improvement. Therein, the authors suggested that the Pd1 site contributes efficient activation of C=O group while Pd-Pd site boosts the dissociation of H₂ molecules to H atoms, according to the ATR experimental results and DFT calculations. This work might be interesting, while there are still number of issues remaining unclear.

1. The major issue is the activity determination. The authors showed that Pd1+NPs/TiO₂ samples are about 3.2 times higher than Pd/C, and 2 times higher than PdNPs/TiO₂. This can be very questionable, since the size of Pd particles in these three samples are very different, such small activity difference might be mainly due to the changes of Pd dispersions in these three samples. If this is the case, the so-called synergy between Pd1 single atoms and NPs might not be true. Therefore, it would be highly useful, if the Pd dispersions of these samples are measured, and the activity was calculated based on the amount of Pd surface atoms in these samples.
2. The second key issue is that stability of Pd1 single atoms. In the liquid phase hydrogenation reaction, the Pd1 atoms might not be stable, they could aggregates to nanoparticles, under reaction conditions. They authors should verify it with the used Pd1/TiO₂ SAC sample.
3. The third issue is about the ATR measurements. What is the Pd/TiO₂ sample measured there? It can be more informative to provide the results of adsorption of MAP on TiO₂, Pd1/TiO₂ and PdNPs/TiO₂, could you see the C=O vibrational shift on Pd1 and Pd NPs? Are the shift different on these two surfaces? This reviewers also feels that MAP could also strongly adsorbs on the TiO₂ support.
4. Another minor issue: the quality of TEM images for Pd1/TiO₂ and PdNPs/TiO₂ in Figures S7 and S8 are very poor, the resolutions of these images are need to be improved.

Reviewer #3 (Remarks to the Author):

This work focuses on the hydrogenation of ketones and aldehydes to alcohols on TiO₂ supported Pd catalysts. The main argument is that there is synergy between the Pd single-atom and the Pd nanoparticle sites: the nanoparticles dissociate H₂ and the H adatoms spillover to the Pd single atoms where they hydrogenate the aldehydes. Experiments thus show higher activity for the catalyst which contains both types of sites. Theory also supports this conclusion: density functional theory calculations show that H₂ dissociation exhibits lower barrier on extended Pd surface sites than Pd1/TiO₂, and a higher binding energy of C₈H₈O (a representative aldehyde) on the Pd single site compared to extended Pd.

The work is very interesting but before it can be considered for publication, the following issues should

be addressed.

The main issue relates to the absence of detailed measurements on the dispersion and number of active sites, which makes it difficult to draw comparisons between the activity of different materials.

In particular, the authors have measured the surface area of mesoporous TiO₂ via BET experiments (lines 87-88 of manuscript and 64-67 of the SI), and they find the Pd loading (4.2 wt%) on the Pd(1+NPs)/TiO₂ from ICP-AES (line 52 in the SI). However, I am not that clear about the number of Pd active sites which they use to deduce the TOF reported in line 118. The authors claim that they employed CO adsorption to measure the dispersion (lines 68-73 of the SI), but I was unable to find the results of these measurements in the main manuscript or in the SI.

This is important, because, unless the number of active sites of the catalytic samples investigated is the same, any comparison is dubious. Comparing the conversion of Pd/C and Pd(1+NPs)/TiO₂ in Figure 2d, does not seem to be a fair, like-for-like comparison. Similarly, it is stated that, in order to get the single atom catalysts Pd₁/TiO₂, the authors reduced the loading to 1% (lines 137-138 of the manuscript). Thus a 4.2x reduction of the loading, from 4.2% to 1%, leads to a 2.3x reduction in the reactivity (57% reduction; line 156). One could argue that this is due to the reduction in the number of active sites.

The main argument of the paper, about the synergy of Pd atoms and nanoparticles is certainly interesting and plausible, but it has to be better substantiated with detailed measurements of the dispersion and the number of active sites.

Moreover, I was wondering: for the Pd₁/TiO₂, is there any risk of these single atoms undergoing sintering while the reaction takes place? If so, this would lead to catalyst deactivation, which was not discussed in the paper.

With regard to the theory (DFT calculations), more details are needed in the description of the methods. The authors should clearly state: how large the simulation cell was (it seems they use 5x5 supercell for Pd, but it is not clear what they do for TiO₂). Which structures were considered for the TiO₂ (anatase or rutile), and which termination(s) for the surface(s). Also, would the PBE functional be good enough, given that self interaction error can be prominent for oxide structures? Finally, vdW interactions (dispersion/London forces) were not included in these calculations, but are known to affect the binding energies of oxygenates. Can the authors comment on this?

Response to the Reviewers' comments

Reviewer #1:

The manuscript from Li et al., describes the development of synergistic catalysts that comprise of isolated Pd sites and Pd NPs on a TiO₂ surface and they propose a mechanism where ketones or aldehydes are bound by the single Pd sites and activated towards reduction with H₂ which is bound and cleaved by the Pd NPs.

Response: Thank you very much for your positive judgment, careful review and valuable comments on our work. Based on your comments, an improved revision has been presented for your consideration.

The introduction appears somewhat limited in scope and should either be broadened to include traditional heterogeneous catalysis as that is stated in the abstract as one of the goals to replace Pd/C. In addition, the reference material should be revised to substantiate and validate the claims. Ref 3 covers MOFs for selectivity of C=O reduction over C=C bonds and photochemical production of Pd NPs. Ref 4 concerns the number of isomers produced on reduction of hexanone which I don't necessarily consider by-products, just low selectivity.

Response: Thanks for your valuable comment! In the revision, we have reorganized the first paragraph and updated the reference materials. We discussed the commercial catalysts and their disadvantages for ketones or aldehydes hydrogenation (Please see Line 6-16 in Paragraph 1 on Page 2 and the Ref. 7-11).

The characterization of the catalysts appears thorough but when looking at the HAADF-STEM image in Fig 2b, it is hard for my untrained eye to definitively see that the circles are definitely single Pd atoms. I looked at work by Tao Zhang in this area (see Fig 1 in Nat. Chem., 3, 634 & in Accounts in Chemical Research, 2013, 46, 8, 1740) and they show a clear difference in the image which is lacking in this figure. Due to this resolution difference, how do the authors rationalize the single atom Pd's in circle and the Pd NPs the boxes? Again, in the work of Zhang these seem significantly different. Do the authors observe any Pd rafts?

Response: Thanks a lot for your careful review. For the HAADF-STEM images, the contrast is greatly effected by the weight of atoms. Pd (106 g/mol) is much lighter than Pt (195 g/mol) and closer to Ti atom (48 g/mol). Therefore, we chose a small particle (~2 nm) to present the adjacent single-atom Pd at the same time. In the revision, (1) we have improved the image by a more experienced technician (Please see updated Figure 2b), where the single atom Pd sites are lighter than the first version of manuscript; (2) we have added images under different magnifications at various sites to show the Pd NPs and Pd single-atoms (Please see the updated Figure S3 and corresponding explanation).

In addition, by carefully observing, we also find a few Pd rafts. Figure 2b and S3b shows a typical one assembled by seven Pd atoms (highlighted by the yellow circle). However, the frequency for Pd rafts is much lower than that of Pd single atoms.

The work to make Pd single atoms and Pd NPs independently is a good way to validate the catalyst and the EXAFS work should be described a little more comprehensively in terms of why the PdO and Pd foil were also analyzed and why those are significant and why they guided the characterization.

Response: Thanks for your good advice. In the revision, we have added more discussion on the

EXAFS work (Please see Line 2-11 on Page 6). Briefly herein, (1) the reference of Pd foil is used to validate the metallic Pd NPs; (2) the reference of PdO is used to confirm the oxidized Pd is originated from single-atom Pd. As we known, Pd-O and Pd-O-Pd coordination will appear simultaneously in the EXAFS spectra of PdO. In the EXAFS spectrum of Pd_{1+NPs}/TiO₂ catalyst, the Pd-O-Pd coordination disappears compared to PdO reference, suggesting the Pd-O coordination is originated from single-atom Pd.

In Figure 3f the caption states that the amount of Pd to substrate is maintained constant. The details for this and the amounts and conditions should be explicitly provided to allow the reader to qualify the results.

Response: Many thanks! In the revision, we have given an explicit description (Please see bottom Line 4-8 on Page 9). The amount of Pd to the substrate is 0.2 mol%. For example, as for one pot reaction with 1 mmol substrate, 5 mg of 4.2 wt.% Pd_{1+NPs}/TiO₂ catalyst powder was added. When we use 1.0 wt.% Pd₁/TiO₂ as catalyst, 21 mg of powder was used.

Table 1 shows some very impressive results. However, no supporting data is provided. I would need to see spectra and an in-depth description of the analytical process to validate this work. I would also like to know if the FID has been calibrated, while more accurate than TIC knowing this is accurate would be appreciated. While the authors talk through the table extensively (this could be shortened) there is no real substance to this section and without the data it's hard to say whether these results are accurate.

Response: Thanks a lot for your careful review. In the revision, we have added all the GC spectra for Table 1 (as shown in Figure S12-19) based on FID. In addition, to make sure the products, we also added some reference spectra with known substance (Please see Figure S13 and 18-20). In this work, we didn't calibrate the FID because our FID responses (peak areas) were well linear to the concentration of the organic analytes (Please see Figure S24).

By the way, our GC was not equipped TIC detector, so we didn't analyze the reaction by TIC.

Have these catalysts been reused and recycled in other experiments or upon completion have the authors added more substrate to test the limits of catalyst use? This should be added to the manuscript.

Response: Thanks for your good advice! In the revision, we have added the experiments of (1) 5 recycled usage of catalysts and (2) increasing the 5 times' substrate to test the limits of catalysts use (Please see the first paragraph on Page 17, Figure S21 and S22). The Pd_{1+NPs}/TiO₂ catalyst exhibit acceptable stability to act as a reusable catalyst for ketone/aldehydes hydrogenation.

Have any other surfaces or supports been investigated? If not why only TiO₂?

Response: Great thanks for your valuable remind! We also tried some other metal oxide supports (Al₂O₃, NiO, Fe₂O₃, Mn₃O₄ and ZnO) with same synthesis method. To our surprise, we found that all of their activities were negligible (Please see Figure S23). This introduces an very interesting issue. However, it can not be make clear in the current work, and we will systematically study the support-dependent activity in the future as a separated project. In the revision, we also give some description about this finding (Please see the last paragraph on Page 17).

Thanks, Andrew Sutton, Los Alamos National Lab.

Response: The authors thank Prof. Sutton again for the careful review and meaningful comments on our manuscript.

Reviewer #2:

In this work, the authors reported that coexistence of Pd₁ single atoms and NPs on TiO₂ support would induce the synergistic function for hydrogenation of ketone/aldehydes up to two times of activity improvement. Therein, the authors suggested that the Pd₁ site contributes efficient activation of C=O group while Pd-Pd site boosts the dissociation of H₂ molecules to H atoms, according to the ATR experimental results and DFT calculations. This work might be interesting, while there are still number of issues remaining unclear.

Response: Thank you very much for your careful review, positive comments, and valuable advice on our work. Based on your comments, an improved revision has been presented for your consideration.

1. The major issue is the activity determination. The authors showed that Pd_{1+NPs}/TiO₂ samples are about 3.2 times higher than Pd/C, and 2 times higher than Pd_{NPs}/TiO₂. This can be very questionable, since the size of Pd particles in these three samples are very different, such small activity difference might be mainly due to the changes of Pd dispersions in these three samples. If this is the case, the so-called synergy between Pd₁ single atoms and NPs might not be true. Therefore, it would be highly useful, if the Pd dispersions of these samples are measured, and the activity was calculated based on the amount of Pd surface atoms in these samples.

Response: Thanks a lot for your valuable advice! In the revision, we have measured the Pd dispersion for commercial Pd/C, Pd₁/TiO₂, Pd_{1+NPs}/TiO₂ and Pd_{NPs}/TiO₂ catalysts and the corresponding dispersion was 32.8%, 67.0%, 17.7% and 6.2% , respectively (Please see the updated Table S2).

Consequently, the turnover frequency (TOF) values based on Pd dispersion were 1218, 645, 4985 and 4565 h⁻¹ for Pd/C, Pd₁/TiO₂, Pd_{1+NPs}/TiO₂ and Pd_{NPs}/TiO₂ catalysts, respectively. (1) The TOF value of Pd_{1+NPs}/TiO₂ is 4.1 times of commercial Pd/C, indicating the advantages of our catalyst. (2) The TOF value of Pd_{1+NPs}/TiO₂ is 7.7 times of Pd₁/TiO₂ catalyst, indicating the crucial role of Pd NPs in high activity for ketone/aldehydes' hydrogenation. (3) The TOF of Pd_{NPs}/TiO₂ is close to that of Pd_{1+NPs}/TiO₂, further confirming the key role of Pd NPs in ketone/aldehydes hydrogenation. The similar TOF values of Pd_{1+NPs}/TiO₂ and Pd_{NPs}/TiO₂ catalysts also suggest the H₂ dissociation is the rate-determining step and the C=O group can be efficiently adsorbed on both Pd NPs and Pd₁ sites.

In the revision, we have added the corresponding discussion (Please first paragraph on Page 11).

2. The second key issue is that stability of Pd₁ single atoms. In the liquid phase hydrogenation reaction, the Pd₁ atoms might not be stable, they could aggregate to nanoparticles, under reaction conditions. They authors should verify it with the used Pd₁/TiO₂ SAC sample.

Response: Many thanks for your good comment! In the revision, we have added the AC-HAADF-STEM images of used Pd₁/TiO₂ SAC sample and given some description (Please see Figure S11 and Line 3-9 on Page 10). Compared with fresh Pd₁/TiO₂ SAC sample (Figure S7), only a few Pd single-atoms aggregate to rafts or clusters (~ 1 nm) while no Pd nanoparticle appear in the used sample, indicating that Pd single atoms are stable under mild hydrogenation conditions.

3. The third issue is about the ATR measurements. What is the Pd/TiO₂ sample measured there? It can be more informative to provide the results of adsorption of MAP on TiO₂, Pd₁/TiO₂ and

$\text{Pd}_{\text{NPs}}/\text{TiO}_2$, could you see the C=O vibrational shift on Pd_1 and Pd NPs? Is the shift different on these two surfaces? This reviewer also feels that MAP could also strongly adsorb on the TiO_2 support.

Response: Thanks a lot for your comments! The ATR result in Figure 4e is from $\text{Pd}_{1+\text{NPs}}/\text{TiO}_2$ sample. In the revision, we have added the ATR measurements for adsorption of MAP on TiO_2 , Pd_1/TiO_2 and $\text{Pd}_{\text{NPs}}/\text{TiO}_2$, and modified corresponding discussion on this part (Please see the last two lines on Page 12 and the first 3 lines on Page 13). As shown in Figure 4e, the MAP adsorption on Pd_1/TiO_2 and $\text{Pd}_{\text{NPs}}/\text{TiO}_2$ is similar to $\text{Pd}_{1+\text{NPs}}/\text{TiO}_2$, where we can also observe the C=O bend vibrational shift. However, the MAP adsorption on TiO_2 were too weak to be detected by our facility. The result suggests that C=O adsorption appears efficiently at the sites of both Pd_1 and Pd_{NPs} . Considered the TOFs of Pd_1/TiO_2 , $\text{Pd}_{1+\text{NPs}}/\text{TiO}_2$ and $\text{Pd}_{\text{NPs}}/\text{TiO}_2$ catalysts, we can conclude that (1) H_2 dissociation plays a more important role in determining the activity; (2) Pd_1 dispersion is key to get abundant active sites and therefore high hydrogenation performance (Please see the last paragraph on Page 14).

4. Another minor issue: the quality of TEM images for Pd_1/TiO_2 and $\text{Pd}_{\text{NPs}}/\text{TiO}_2$ in Figures S7 and S8 are very poor, the resolutions of these images are need to be improved.

Response: Thank you very much for your good advice! To improve the quality of TEM images, we have redone them by an aberration-corrected high-angle annular dark field-STEM (AC-HAADF-STEM, please see updated Figure S7, S8 and corresponding explanations), whose resolution reaches 0.08 nm (JEM-ARM200F, JEOL).

Reviewer #3:

This work focuses on the hydrogenation of ketones and aldehydes to alcohols on TiO₂ supported Pd catalysts. The main argument is that there is synergy between the Pd single-atom and the Pd nanoparticle sites: the nanoparticles dissociate H₂ and the H adatoms spillover to the Pd single atoms where they hydrogenate the aldehydes. Experiments thus show higher activity for the catalyst which contains both types of sites. Theory also supports this conclusion: density functional theory calculations show that H₂ dissociation exhibits lower barrier on extended Pd surface sites than Pd₁/TiO₂, and a higher binding energy of C₈H₈O (a representative aldehyde) on the Pd single site compared to extended Pd.

The work is very interesting but before it can be considered for publication, the following issues should be addressed.

Response: The authors thank the reviewer very much for your careful review and positive comments on our work. Based on your valuable comments and advice, we present an improved revision for your consideration.

The main issue relates to the absence of detailed measurements on the dispersion and number of active sites, which makes it difficult to draw comparisons between the activity of different materials.

In particular, the authors have measured the surface area of mesoporous TiO₂ via BET experiments (lines 87-88 of manuscript and 64-67 of the SI), and they find the Pd loading (4.2 wt%) on the Pd_(1+NPs)/TiO₂ from ICP-AES (line 52 in the SI). However, I am not that clear about the number of Pd active sites which they use to deduce the TOF reported in line 118. The authors claim that they employed CO adsorption to measure the dispersion (lines 68-73 of the SI), but I was unable to find the results of these measurements in the main manuscript or in the SI.

Response: Thanks for your valuable comment! In the revision, we have added the Pd dispersion for commercial Pd/C, Pd₁/TiO₂, Pd_(1+NPs)/TiO₂ and Pd_{NPs}/TiO₂ catalysts and the corresponding dispersion was 32.8%, 67.0%, 17.7% and 6.2% , respectively (Please see the updated Table S2). The number of active sites were calculated based on the amount of Pd usage (0.002 mmol) and Pd dispersion. The measurement details has been added in the supporting information (Please the last paragraph on Page 3 and first two paragraph on Page 4 in SI).

This is important, because, unless the number of active sites of the catalytic samples investigated is the same, any comparison is dubious. Comparing the conversion of Pd/C and Pd_(1+NPs)/TiO₂ in Figure 2d, does not seem to be a fair, like-for-like comparison. Similarly, it is stated that, in order to get the single atom catalysts Pd₁/TiO₂, the authors reduced the loading to 1% (lines 137-138 of the manuscript). Thus a 4.2x reduction of the loading, from 4.2% to 1%, leads to a 2.3x reduction in the reactivity (57% reduction; line 156). One could argue that this is due to the reduction in the number of active sites.

Response: Many thanks for your good comment! The loading of Pd₁/TiO₂ was indeed 4.2x reduction. During catalytic evaluation, the amount of Pd to the substrate is kept as 0.2 mol% for all catalysts. For example, as for one pot reaction with 1 mmol substrate, 5 mg of 4.2 wt.% Pd_(1+NPs)/TiO₂ catalyst powder was added. When we use 1.0 wt.% Pd₁/TiO₂ as catalyst , 21 mg of powder was used.

For meaningful comparison, the activities were evaluated with turnover frequency (TOF) based on Pd dispersion (Please see the first paragraph on Page 11). Consequently, the turnover frequency

(TOF) values based on Pd dispersion were 1218, 645, 4985 and 4565 h⁻¹ for Pd/C, Pd₁/TiO₂, Pd_{1+NPs}/TiO₂ and Pd_{NPs}/TiO₂ catalysts, respectively. (1) The TOF value of Pd_{1+NPs}/TiO₂ is 4.1 times of commercial Pd/C, indicating the advantages of our catalyst. (2) The TOF value of Pd_{1+NPs}/TiO₂ is 7.7 times of Pd₁/TiO₂ catalyst, indicating the crucial role of Pd NPs in getting high activity for ketone/aldehydes' hydrogenation.

The main argument of the paper, about the synergy of Pd atoms and nanoparticles is certainly interesting and plausible, but it has to be better substantiated with detailed measurements of the dispersion and the number of active sites.

Response: Thank you! In the revision, we have measured the Pd dispersions of Pd_{1+NPs}/TiO₂, Pd_{NPs}/TiO₂ and commercial Pd/C catalysts by CO pulse chemisorption. As for the Pd₁/TiO₂, the CO pulse chemisorption is inapplicable to determine the dispersion of Pd because the weak binding of CO on Pd₁ leads to desorption of CO. Similar result has been studied for single-atom Pt in both Pt/Cu alloys by Flytzani-Stephanopoulos' group (*J. Am. Chem. Soc.* **2016**, *138*, 6396-6399) and in TiO₂ support by Christopher' group (*Nat. Mater.* **2019**, *18*, 746-751). According to Pratsinis et al.' work (*Appl. Catal. B: Environ.* **2018**, *226*, 127-134), we estimated the Pd dispersion as 67% for 1% Pd₁/TiO₂ by leaching the surface Pd with a mixture of H₂O₂ and HCl solution.

In the supporting information, we have added more details about the measurement (the last paragraph on Page 3 and first two paragraph on Page 4 in Supporting Information). The number of active sites were calculated based on the amount of Pd usage (0.002 mmol) and Pd dispersion.

Moreover, I was wondering: for the Pd₁/TiO₂, is there any risk of these single atoms undergoing sintering while the reaction takes place? If so, this would lead to catalyst deactivation, which was not discussed in the paper.

Response: Thanks a lot! In the revision, we have added the HAADF-STEM images of used Pd₁/TiO₂ SAC sample (under 1 hour's continuous reaction) and given some description (Please see Figure S11 and Line 3-9 on Page 10). Compared with fresh Pd₁/TiO₂ SAC sample (Figure S7), only a few Pd single-atoms aggregate to rafts or clusters (~ 1 nm) while no Pd nanoparticle appear in the used sample, indicating that the stability of Pd single atoms are acceptable to discuss their activity under mild hydrogenation conditions.

With regard to the theory (DFT calculations), more details are needed in the description of the methods. The authors should clearly state: how large the simulation cell was (it seems they use 5x5 supercell for Pd, but it is not clear what they do for TiO₂). Which structures were considered for the TiO₂ (anatase or rutile), and which termination(s) for the surface(s)? Also, would the PBE functional be good enough, given that self-interaction error can be prominent for oxide structures? Finally, vdW interactions (dispersion/London forces) were not included in these calculations, but are known to affect the binding energies of oxygenates. Can the authors comment on this?

Response: Thanks a lot for your remind! In the revision, we have added more description for the DFT calculations (Please see the second paragraph on Page 6 in Supporting Information). According to the XRD pattern (Figure S1c), anatase TiO₂ crystal was chosen to cleave the stable TiO₂ (110) surface and facet-centered cubic (FCC) Pd crystal was used to cleave Pd (111) surface. The model of (3x2) supercell slab (Ti₃₆O₇₂) including six atomic layers was adopted for anatase TiO₂ (110) surface. The termination of TiO₂ (110) surface contains the 4-fold coordinated Ti (Ti_{4c})

and 2-fold coordinated O (O_{2c}). The model of (4×4) supercell slab (Pd_{96}) including six atomic layers was adopted for FCC Pd (110) surface.

In addition, we chose PBE functional to describe the electron–electron interaction because we found it has been successful used to deal with TiO_2 (*Phys. Chem. Chem. Phys.* **2014**, *16*, 21446-21451; *Nat. Mater.* **2019**, *18*, 746-751).

As for the vdW interactions, just as you said, it was indeed not considered in these calculations. The reasons are (1) the coverage of oxygenates is low (one molecule in (4×4) Pd (111) and (3×2) TiO_2 (110) supercell slabs, respectively) and the distance between two C_8H_8O molecules is long, which reduces the vdW interaction of oxygenates; (2) the effect of vdW interactions is generally lower than 0.1 eV, which is much lower than the chemical adsorption energies of C_8H_8O molecule on Pd (111) (-0.66 eV) and Pd_1/TiO_2 (110) (-1.31 eV). In the revision, we have added the corresponding description (Please see Line 3-9 in Paragraph 1 on Page 12).

Reviewers' comments:

Reviewer #2 (Remarks to the Author):

In this revised manuscript, the authors have addressed most of the comments raised previously. One particular result from ATR supports that the MAP substrate could adsorb on the Pd atoms and Pd NPs. However, the major issues related with the activity still remained:

1. Their new data showed that the activity difference between Pd1+NPs/TiO₂ and PdNPs/TiO₂ was so little (4985 vs 4565 h⁻¹). Such trivial difference might be only attributed to the Pd particle size effect instead of the synergy effect between Pd atoms and NPs claimed by the authors, since the Pd dispersions in these two samples are different. In literature, it has been shown particle size variation of could induce an even large activity changes (RSC Adv., 2016, 6, 75541-75551; J. Am. Chem. Soc. 2006128144510-4511). How do the authors can rule out the Pd particle size effect? What are the TOFs for the ~2 nm or 3 nm Pd particles on TiO₂ instead of ~10 nm reported here?
2. The authors showed that there is strong support effect, and Pd NPs supported on other oxides such as Al₂O₃, NiO, Fe₂O₃, Mn₃O₄ and ZnO had no activity at all. Therefore, it will be not fair to compare their activity of Pd1+NPs/TiO₂ with Pd/C.

Reviewer #3 (Remarks to the Author):

The revised manuscript has partly addressed my previous comments. Questions remain about the applicability of the functional used for the DFT calculations. The authors note in the rebuttal that they "chose [the] PBE functional to describe the electron–electron interaction because [they] found it has been successful[ly] used to deal with TiO₂ (Phys. Chem. Chem. Phys. 2014, 16, 21446-21451; Nat. Mater. 2019, 18, 746-751)". Yet, both papers cited actually adopted the DFT+U method and they mention explicitly the value of the U parameter they have used. Can the authors clarify whether they use DFT+U or the "plain" PBE functional, which would probably be rather inadequate for TiO₂? In addition, the arguments they present for not considering vdW interactions are not convincing. I am not sure how they justify that "the effect of vdW interactions is generally lower than 0.1 eV". Differences between the binding energies predicted by PBE and vdW functionals (or post-SCF correction methods) can certainly be much higher than 0.1, for instance, differences on the order of 0.5 eV have been reported for oxygenates binding on gold (J. Phys. Chem. C 2017, 121, 50: 27905-27914). The other two arguments (low coverages of oxygenates; long distance between adsorbates), are clearly irrelevant, since vdW interactions can be exerted between the adsorbates and the TiO₂ surface.

Moreover, there are several typos in the newly introduced text. Aside from grammar or spelling errors (e.g. "facet-center cubic" should be "face-centered cubic" on SI page 6), the authors should carefully check references to figures, tables, etc., for instance, in the SI, on line 238, Fig. 12 is referenced, whereas this should be Fig. 11.

Response to the reviewers' comments:

Reviewer 2:

In this revised manuscript, the authors have addressed most of the comments raised previously. One particular result from ATR supports that the MAP substrate could adsorb on the Pd₁ atoms and Pd NPs. However, the major issues related with the activity still remained:

Response: Thank you very much for the review again! These valuable comments are gratefully appreciated. In the revision, we further clarify the synergy effect between Pd atoms and NPs with a tandem working mechanism.

1. Their new data showed that the activity difference between Pd_{1+NPs}/TiO₂ and Pd_{NPs}/TiO₂ was so little (4362 vs 4565 h⁻¹). Such trivial difference might be only attributed to the Pd particle size effect instead of the synergy effect between Pd atoms and NPs claimed by the authors, since the Pd dispersions in these two samples are different. In literature, it has been shown particle size variation of could induce an even large activity changes (RSC Adv., 2016, 6, 75541-75551; J. Am. Chem. Soc. 2006, 128(14), 4510-4511). How do the authors can rule out the Pd particle size effect? What are the TOFs for the ~2 nm or 3 nm Pd particles on TiO₂ instead of ~10 nm reported here?

Response: Thank you for bringing this up! We agree that the TOF difference between Pd_{1+NPs}/TiO₂ and Pd_{NPs}/TiO₂ is small. As you commented, from the perspective of the TOFs, we can't really draw such a conclusion on the synergy function of Pd₁ atoms and NPs. If we shift the perspective to the hydrogenation performance (reactivity), we might observe the synergy function of Pd₁ atoms and NPs (Please see the following analysis around Table R1 and Figure R1). In combination with the proposed working mechanism, we might further understand their synergy function (Please see the following discussion around Figure 1).

As shown in Table R1, although Pd_{NPs}/TiO₂ had high TOFs value, it presented a low-scale reactivity due to the low metal dispersion. When the metal dispersion was increased to atomic dispersion, Pd₁/TiO₂ also presented a low-scale reactivity due to the low TOFs value. Therefore, there was a negative correlation between activity and dispersion. This is consistent with the result of Crook et al.'s work (J. Am. Chem. Soc. 2006, 128(14), 4510-4511), in which the surface atom normalized TOF values decreased with the particle size.

Table R1: The activity and reactivity data and the contribution response analysis.

Evaluation	Catalysts			Contribution Response	
	Pd ₁ /TiO ₂	Pd _{NPs} /TiO ₂	Pd _{1+NPs} /TiO ₂	Pd ₁	Pd _{NPs}
Activity (TOFs, h ⁻¹)	Low 645	High 4565	High 4362		√
Reactivity (Conversion at 15 min)	Medium 20%	Low 14%	High 41%	√	√

The trade-off between activity and dispersion was achieved by coexistence of Pd₁ and Pd_{NPs} in Pd_{1+NPs}/TiO₂ catalyst, which exhibited much higher reactivity than both Pd₁/TiO₂ and Pd_{NPs}/TiO₂ catalysts. In addition, we studied a 2.8 % metal loaded Pd_{1+NPs}/TiO₂ catalyst with less and smaller Pd NPs (Figure R1a-c) than 4.8 % Pd_{1+NPs}/TiO₂ catalyst. The TOFs of 2.8 % Pd_{1+NPs}/TiO₂ was measured as 1341 h⁻¹ (Figure R1d), much lower than that of 4.8 % Pd_{1+NPs}/TiO₂ but much higher than that of Pd₁/TiO₂ (645 h⁻¹). To some extent, this finding is also consistent with the result of Crook et al.'s work.

Figure R1: The TEM (a), HAADF-STEM images (b and c) of 2.8 % Pd_{1+NPs}/TiO₂ catalyst, and the corresponding 4-methylacetophene (MAP) conversion plots (d) under 1 atm H₂ pressure at 25 °C

In the revision, synergy function of Pd₁ atoms and NPs was further clarified through a tandem working mechanism of Pd₁ atoms and NPs. The part of mechanism discussion was rewritten (as shown in the last paragraph on Page 14 to the first paragraph on Page 15, and the Ref. 52 and 53). According to DFT results, H₂ will dissociate preferentially on metallic Pd nanoparticles and ketone/aldehydes molecules can be adsorbed at Pd₁ sites. As shown in Figure 1, the H₂ molecules are dissociated into H atoms on Pd NPs with fast kinetics. The spillover of H atoms from Pd NPs migrates to the ketone/aldehydes-adsorbed Pd₁ sites. The hydrogenation was finally completed at Pd₁ sites. Herein, both Pd NPs and Pd₁ atoms are key to high hydrogenation performance. In terms of the Pd_{NPs}/TiO₂ with high activity, the nature of low dispersion caused a low hydrogenation exhibition. While, although atomically Pd₁/TiO₂ catalyst possessed the highest dispersion, the high H₂ dissociate barrier induced low activity still made it present a low-level hydrogenation performance. The tandem working mechanism dissolved the disadvantages of both single-atom Pd₁ site (low activity) and Pd NPs (low dispersion) synergistically. According to Crook et al.'s work (J. Am. Chem. Soc. 2006, 128, 4510-4511) and working mechanism of Pd₁ atoms and NPs, the size of Pd NPs and ratio of Pd NPs/Pd₁ sites would affect the hydrogenation performance of Pd_{1+NPs}/TiO₂ catalyst.

Figure 1: The model and working mechanism of Pd_{1+NPs}/TiO₂ synergistic catalyst for ketone/aldehydes hydrogenation.

2. The authors showed that there is strong support effect, and Pd NPs supported on other oxides such as Al₂O₃, NiO, Fe₂O₃, Mn₃O₄ and ZnO had no activity at all. Therefore, it will be not fair to

compare their activity of Pd_{1+NPs}/TiO₂ with Pd/C.

Response: Thank you for the comment. We realize that it is unsuitable to claim Pd_{1+NPs}/TiO₂ as a “commercial Pd/C-replaceable catalyst” because Pd/C catalyst works efficiently in other fields. We modified the text in the revision accordingly. To position the Pd_{1+NPs}/TiO₂ in terms of ketone/aldehydes hydrogenation performance, the result of Pd/C was still retained as a benchmark catalyst. Nevertheless, as for all the discussion referred to Pd/C catalyst, the comparison was just focused on the performance in the ketone/aldehydes hydrogenation. The discussion has also been amended (Please the last 2 lines on Page 7, the first paragraph on Page 8).

Reviewer 3:

1. The revised manuscript has partly addressed my previous comments. Questions remain about the applicability of the functional used for the DFT calculations. The authors note in the rebuttal that they "chose the PBE functional to describe the electron–electron interaction because they found it has been successfully used to deal with TiO₂ (Phys. Chem. Chem. Phys. 2014, 16, 21446-21451; Nat. Mater. 2019, 18, 746-751)". Yet, both papers cited actually adopted the DFT+U method and they mention explicitly the value of the U parameter they have used. Can the authors clarify whether they use DFT+U or the "plain" PBE functional, which would probably be rather inadequate for TiO₂? In addition, the arguments they present for not considering vdW interactions are not convincing. I am not sure how they justify that "the effect of vdW interactions is generally lower than 0.1 eV". Differences between the binding energies predicted by PBE and vdW functionals (or post-SCF correction methods) can certainly be much higher than 0.1, for instance, differences on the order of 0.5 eV have been reported for oxygenates binding on gold (J. Phys. Chem. C 2017, 121, 50: 27905-27914). The other two arguments (low coverages of oxygenates; long distance between adsorbates), are clearly irrelevant, since vdW interactions can be exerted between the adsorbates and the TiO₂ surface.

Response: Thank you very much! Based on your valuable comments, we further improved our manuscript. In the revision, we have modified the DFT study, in which we (1) considered the vdW interactions for all calculations and (2) used DFT+U method to deal with TiO₂-relative calculations. The Grimme method was adopted for DFT-D correction. The U value was set as 7 eV according to (Phys. Chem. Chem. Phys. 2014, 16, 21446-21451).

Table S3: The adsorption energies of C₈H₈O and H₂ molecules on the surface of Pd (111) and Pd₁/TiO₂(110) based on DFT calculation.

System	$\Delta E/eV$	
	C ₈ H ₈ O	H ₂
Pd(111)	-0.34194	-0.08831
Pd ₁ /TiO ₂ (110)	-0.91719	-0.15283

Figure 4j: The relative energy plots (j) of H₂ dissociation on Pd₁/TiO₂ (110) (red) and Pd (black) surfaces

We agree that the vdW interactions indeed affected significantly on the adsorption energy. Thank you for the comment again! In the revision, the updated configurations and results have been shown in Figure 4 and Table S3. The updated relative energy plots of H₂ dissociation has been shown in Figure 4f. We also corrected the corresponding description in the text (Please see

Line 1-7 on Page 12, second paragraph on Page 6 in Supporting Information).

2. Moreover, there are several typos in the newly introduced text. Aside from grammar or spelling errors (e.g. "facet-center cubic" should be "face-centered cubic" on SI page 6), the authors should carefully check references to figures, tables, etc., for instance, in the SI, on line 238, Fig. 12 is referenced, whereas this should be Fig. 11.

Response: Thank you for your comment! In the revision, we have carefully checked the text and corrected the typos and figure/table references etc.

REVIEWERS' COMMENTS:

Reviewer #2 (Remarks to the Author):

In the revised manuscript. The authors have carefully performed additional experiments and addressed the previous issues. The new results further clarified the Synergistic Function of Pt1 and PtNPs. This work not only contributes a highly active catalyst for ketone/aldehydes hydrogenation, but also providing new insight on the cooperative effect between single atoms and nanoparticles which are present in many cases. Therefore this reviewer recommends it for publication.

Reviewer #3 (Remarks to the Author):

Thank you for addressing my comment and it is good to see that the qualitative picture remains the same. One last thing that can be addressed at proofing stage at the author's discretion (no need for another review): consider explicitly stating the level of the DFT-D method (B97-D, judging from the reference), as there are several functionals within Grimme's DFT-D family of methods. Other than that, I think the article can now be accepted for publication.

Response to the reviewers

Reviewer #2:

In the revised manuscript, the authors have carefully performed additional experiments and addressed the previous issues. The new results further clarified the Synergistic Function of Pt₁ and Pt_{NPs}. This work not only contributes a highly active catalyst for ketone/aldehydes hydrogenation, but also providing new insight on the cooperative effect between single atoms and nanoparticles which are present in many cases. Therefore this reviewer recommends it for publication.

Response: Thank you very much for your positive evaluation, all the previous valuable comments and publication recommendation on this work.

Reviewer #3:

Thank you for addressing my comment and it is good to see that the qualitative picture remains the same. One last thing that can be addressed at proofing stage at the author's discretion (no need for another review): consider explicitly stating the level of the DFT-D method (B97-D, judging from the reference), as there are several functionals within Grimme's DFT-D family of methods. Other than that, I think the article can now be accepted for publication.

Response: The authors appreciate the reviewers' careful review and good comment. In the revision, we have stated the level of the DFT-D method to "B97-D" (Please see bottom line 5 on Page 12 and line 4 on Page 24), which was highlighted with yellow background. Finally, we thank the reviewer for all the previous valuable comments and publication recommendation on this work.